# OpenReview forum: "TRUTH DECAY: Quantifying Multi-Turn Sycophancy in Language Models"
_ICLR.cc/2025/Workshop/BuildingTrust — Submitted to BuildingTrust_

### Official Review · Reviewer_DDVF · 2025-02-16
**Review of TRUTH DECAY**

**Rating:** 6
**Confidence:** 5

**Review:**

# Summary
The paper introduces TRUTH DECAY, a multi-turn evaluation benchmark designed to measure sycophancy in Large Language Models (LLMs). It evaluates factual accuracy degradation as models interact with users over multiple conversational turns, using both static follow-up prompts and rationale-based adversarial probes. The evaluation is conducted on TruthfulQA and MMLU-Pro with several models (Claude Haiku, GPT-4o-mini, LLaMA 3.1 8B), showing that sycophancy increases over turns, particularly in subjective domains like philosophy. Two simple mitigation prompts are also tested.

## Strengths

- Addresses an important issue in LLM evaluation: factual degradation and sycophantic alignment in multi-turn dialogues.

- Methodologically sound, combining existing datasets with a multi-turn evaluation structure.

- Useful practical insights on accuracy degradation across domains and the potential for simple prompting strategies to reduce sycophancy.

## Weaknesses

- The novelty is somewhat limited, as multi-turn factual degradation has been explored in prior work (e.g., Laban et al., 2024). This work focuses more specifically on sycophancy, which is a valuable extension but not a major conceptual leap.

- The benchmark is more of a practical evaluation setup than a foundational benchmark, which is fine for a workshop but should be framed as such.

- Results align largely with expectations—sycophancy increases over turns, and subjective domains degrade more than STEM—but the findings are still practically useful.

## Suggested Improvements

- Clarify the positioning of the benchmark as a practical evaluation pipeline rather than a foundational benchmark.

- Acknowledge and situate the work more clearly alongside Laban et al. (2024) and [Scheurer et al.](https://arxiv.org/pdf/2311.07590) to emphasize its contribution as an extension to sycophancy evaluation.

---

### Official Review · Reviewer_xHuN · 2025-02-23

**Rating:** 5
**Confidence:** 4

**Review:**

The paper explores the phenomenon of sycophancy in large language models (LLMs), specifically in multi-turn conversations. It introduces TRUTH DECAY, a benchmark designed to evaluate sycophantic behavior over extended dialogues, where language models must handle iterative user feedback, persuasion, and challenges. The study highlights how LLMs, particularly those trained with reinforcement learning from human feedback (RLHF), can drift toward excessive agreement, sacrificing factual accuracy. The authors test various sycophancy reduction strategies and demonstrate that such biases worsen with multiple exchanges. The paper suggests that current LLMs struggle to maintain objectivity and truthfulness during extended interactions and proposes methods to mitigate these effects in future models.

Strengths
- The TRUTH DECAY benchmark provides a structured and in-depth evaluation of sycophantic behavior in multi-turn conversations, a critical aspect of LLM performance.
- The study highlights how current LLMs, particularly those trained with RLHF, exhibit sycophantic tendencies, shedding light on important model behavior that needs attention.
- The paper not only identifies sycophancy issues but also proposes potential strategies to reduce biases, paving the way for more objective and accurate future models.

Weaknesses
- The TRUTH DECAY benchmark may not capture all nuances of sycophantic behavior, potentially overlooking subtler forms of bias in different contexts.
- The focus on models trained with reinforcement learning from human feedback may limit the generalizability of the findings to other types of model training approaches.
- The proposed sycophancy reduction strategies might not scale effectively to larger, more complex models, potentially requiring significant adjustments or trade-offs in performance.
- The citation form is not appropriate and the table captions are not correctly placed above the tables.

---

### Official Review · Reviewer_7aun · 2025-03-01
**The paper introduces a useful multi-turn sycophancy benchmark but needs better clarity and justification for some claims**

**Rating:** 3
**Confidence:** 3

**Review:**

## Summary

This paper introduces Truth Decay, a new benchmark designed to evaluate sycophantic tendencies in large language models (LLMs) over multi-turn interactions. The authors argue that existing sycophancy evaluation methods primarily focus on single-turn responses and fail to capture the compounding nature of sycophancy in extended conversations. To address this gap, the paper proposes two evaluation strategies: (1) static feedback-based sycophancy, where models are tested with structured follow-up prompts, and (2) rationale-based sycophancy, where incorrect responses are reinforced with persuasive rationales generated by another LLM. Additionally, the paper evaluates sycophancy mitigation techniques through specific anti-sycophancy prompts. The study assesses Claude Haiku, GPT-4o-mini, and Llama 3.1 8B across these settings, demonstrating how models progressively adopt sycophantic behaviors over multiple turns and struggle to maintain factual accuracy.

## Strengths

- Important Problem Addressed: The paper highlights a significant limitation of current LLMs—their susceptibility to sycophantic behavior, which can compromise reliability in high-stakes applications.
- Multi-Turn Evaluation: Unlike existing single-turn sycophancy tests, this study systematically examines how model responses evolve over extended interactions, revealing the compounding nature of sycophancy.
- Diverse Testing Approaches: The inclusion of both static follow-ups and rationale-based interventions provides a nuanced perspective on how models internalize and propagate incorrect information.
- Discussion on Anchoring Effects: Section 5.4 presents an interesting discussion on how models may exhibit anchoring biases, making it difficult to revert to factual correctness after an initial incorrect response.

## Weaknesses

- Writing Clarity and Organization:
  - The paper uses incorrect citation formatting (e.g., should use `\citep{}` when references are not integral to the sentence) and quoation marks (e.g., should use ``latex quotation marks'')
  - The abstract does not clearly outline the paper's contributions and key findings.
  - The introduction effectively presents the problem but does not sufficiently describe the proposed evaluation approach.
  - The related work section lacks a more explicit comparison with existing multi-turn benchmarking efforts.
  - Figures are not consistently referenced within the text, making it difficult  to follow the discussion.

- Limited Evaluation Scope: The experimental results are constrained to three models (Claude Haiku, GPT-4o-mini, Llama 3.1 8B), which may not generalize to other state-of-the-art models with stronger anti-sycophancy safeguards. The authors should also specify which specific version of Claude Haiku was used.

- Weak Justification for Some Claims:
  - Section 5.1 claims that sycophantic tendencies are already present in single-step dialogues and worsen in multi-turn settings but does not clearly indicate which results support this claim.
  - Some key results supporting the analysis are only in the appendix, without clear references in the main text.

## Overall Evaluation

The paper addresses an important issue and presents a novel approach to studying sycophancy in LLMs beyond single-turn interactions. The multi-turn evaluation paradigm is a valuable addition to existing benchmarks, and the experiments provide useful insights into how sycophancy compounds over repeated interactions. However, the paper would benefit from clearer writing, better justification of claims, and a stronger discussion of its relation to existing multi-turn benchmarks. Additionally, providing a more comprehensive evaluation with a broader range of models would strengthen its contributions. Overall, while the paper presents valuable work, its clarity and rigor need improvement before publication.

---

### Decision · Program_Chairs · 2025-03-04

Reject